# A biomimetic chiral-driven ionic gate constructed by pillar[6]arene-based host–guest systems

Yue Sun[1], Fan Zhang[1], Jiaxin Quan[1], Fei Zhu[1], Wei Hong[1], Junkai Ma[1], Huan Pang[1], Yao Sun[1], Demei Tian[1] & Haibing Li[1]

Inspired by glucose-sensitive ion channels, herein we describe a biomimetic glucose-enantiomer-driven ion gate via the introduction of the chiral pillar[6]arene-based host–guest systems into the artificial nanochannels. The chiral nanochannels show a high chiral-driven ionic gate for glucose enantiomers and can be switched "off" by D-glucose and be switched "on" by L-glucose. Remarkably, the chiral nanochannel also exhibited a good reversibility toward glucose enantiomers. Further research indicates that the switching behaviors differed due to the differences in binding strength between chiral pillar[6]arene and glucose enantiomers, which can lead to the different surface charge within nanochannel. Given these promising results, the studies of chiral-driven ion gates may not only give interesting insight for the research of biological and pathological processes caused by glucose-sensitive ion channels, but also help to understand the origin of the high stereoselectivity in life systems.

[1] Key Laboratory of Pesticide and Chemical Biology (CCNU), Ministry of Education, College of Chemistry, Central China Normal University, 430079 Wuhan, People's Republic of China. Correspondence and requests for materials should be addressed to H.L. (email: lhbing@mail.ccnu.edu.cn)

The gating behavior of glucose-sensitive ion channels is critical for the maintenance of glucose homeostasis[1–3]. The opening and closing of the glucose-sensitive ion channels are involved in glucose uptake into skeletal muscle and release from the liver. The loss- or gain-of-function gating in glucose-sensitive ion channels has profound effects, giving rise to neonatal diabetes[4,5]. Recent studies showed that the gating behavior of glucose-sensitive ion channels can be mediated by external stimulus[6,7]. Chirality is a fundamental character of living matter and glucose is a chiral molecule[8]. Accumulating evidence has indicated that most biological and physiological processes involve intermolecular interactions that are mediated by chirality[9]. These messages triggered our thought: whether the chirality regulates the gating behavior of glucose-sensitive ion channels?

The considerable complexity and variability in biological nanochannels severely hinders the practical studies on glucose-sensitive ion gates. To explore various biological processes in vitro, biomimetic strategies have been recently developed[10,11]. Compared with biological channels, solid-state synthetic nanochannels have attracted great interest due to their outstanding mechanical properties and good chemical stability[12]. By incorporating functional groups onto the nanochannels surface, the artificial system can readily present versatile gate properties and respond to diverse environmental stimuli. Recently, we exploited a strategy of introducing host–guest systems into the internal walls of nanochannels for the construction of an artificial ion gate, unitizing the good reversibility and high selectivity of host–guest systems[13–16]. These results prompted us to design a rational receptor with chirality to further study glucose-sensitive ion gate. Pillararenes based host–guest systems are good candidates for designing chiral receptors[17–25]. Not only can they be easily synthesized and modified with multi-chiral groups, but they also possess the appropriate spatial disposition for binding glucose, thereby providing an ideal platform to bridge this gap.

Herein, inspired by biological glucose-sensitive ion gate, we report a simple and effective nanochannel embedded in a chiral pillar[6]arene-based host–guest system to study the glucose-enantiomer-driven ion gate (Fig. 1). Chiral alanine-decorated pillar[6]arene was synthesized by the condensation reaction. Then chiral receptor was installed onto the internal surface of the nanochannel by the host–guest interactions. Through regulating the surface charge, D-glucose (D-Glu) can precisely control ion transport with a gating ratio up to 7, in contrast to L-glucose (L-Glu). Remarkably, the chiral nanochannel exhibited good reversibility toward glucose enantiomers. The switching behaviors differed, which may be explained by the differences in the surface charge within the nanochannel caused by the binding strength between the two complexes. Given these promising results, in-depth studies of the artificial ion gate driven by glucose enantiomers are very useful to not only understand the biological and pathological processes caused by glucose-sensitive ion gate, but also provide a technique to simulate the transport properties and gating behavior of specialized channels.

## Results

**Design and synthesis of chiral pillar[6]arene.** The multiple hydrogen-bonding interactions play a critical role in the glucose-sensitive ion gate[26]. Amino acids, not only occurring as active sites in glucose-sensitive ion channels, but also being excellent hydrogen-bonding donors as the binding sites, were decorated with pillararenes[27–29]. More importantly, the pillararenes structure appending with multiple chiral amino acids may provide a strong chiral environment that serves to benefit chiral ion gate[30]. Thus, they may be used as chiral receptor systems for fabricating chiral gated channel. To prove this idea, we design and synthesize pillar[6]arene modified with L-alanine. The synthetic procedure to prepare the chiral receptors is depicted in Fig. 2; bromine-functionalized pillar[6]arene 2 is synthesized according to the literature[31]. Subsequently, 1 is synthesized by a reduction reaction that reacted with Pd/C in methanol under a hydrogen atmosphere. Then, chiral pillar[6]arene is prepared through condensation and removal of the Boc-protecting group, using 1 as the starting material. The structure and configurations were investigated by NMR (nuclear magnetic resonance), MALDI-TOF MS (matrix assisted laser desorption/ionization mass spectrometry) , and CD (circular dichroism) spectroscopy (see Supplementary Figs. 27, 28, and 29). As expected, the pillar[6]arene not only possesses the alanine CD signal at about 215 nm but also demonstrates the positive Cotton effect at approximately 245 nm and positive at 310 nm for the region of planar chirality (see Supplementary Fig. 30)[32].

**Fabrication of the chiral nanochannel.** With the L-AP6 receptor in hand, we aim to fabricate artificial nanochannel that could investigate the glucose-enantiomer-driven ion gate. However, to achieve the aforementioned, we had to determine the degree of immobilization needed for the chiral receptor. Thus, we elaborately designed the AZO compound (Fig. 3a). The terminal amino of AZO was connected with the nanochannel surface using a classic condensation reaction. Furthermore, AZO has proven to be especially advantageous for introducing pillar[6]arene via host–guest interactions[33]. We primarily attempt to study the interaction between L-AP6 host and AZO guest. [1]H NMR titration experiments are assigned to calculate the association constant ($K_a$) and stoichiometry of this host–guest complex. The $K_a$ value is estimated to be $(503 \pm 1)$ M$^{-1}$ for AZO $\subset$ L-AP6 in the 1:1 complexation mode (see Supplementary Fig. 3). On the basis of these studies, an achiral nanochannel can be readily converted into a chiral biomimetic channel by supramolecular self-assembly. Fig. 3b describes the process of fabricating the

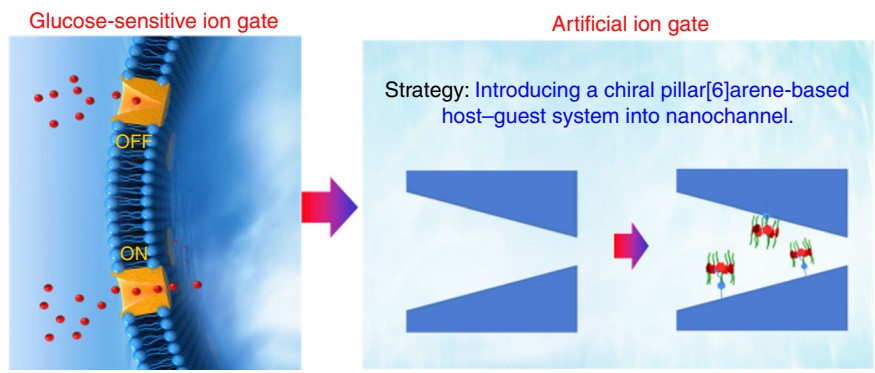

**Fig. 1** Schematic of the design of a biomimetic glucose-enantiomer-driven ion gate using host–guest systems

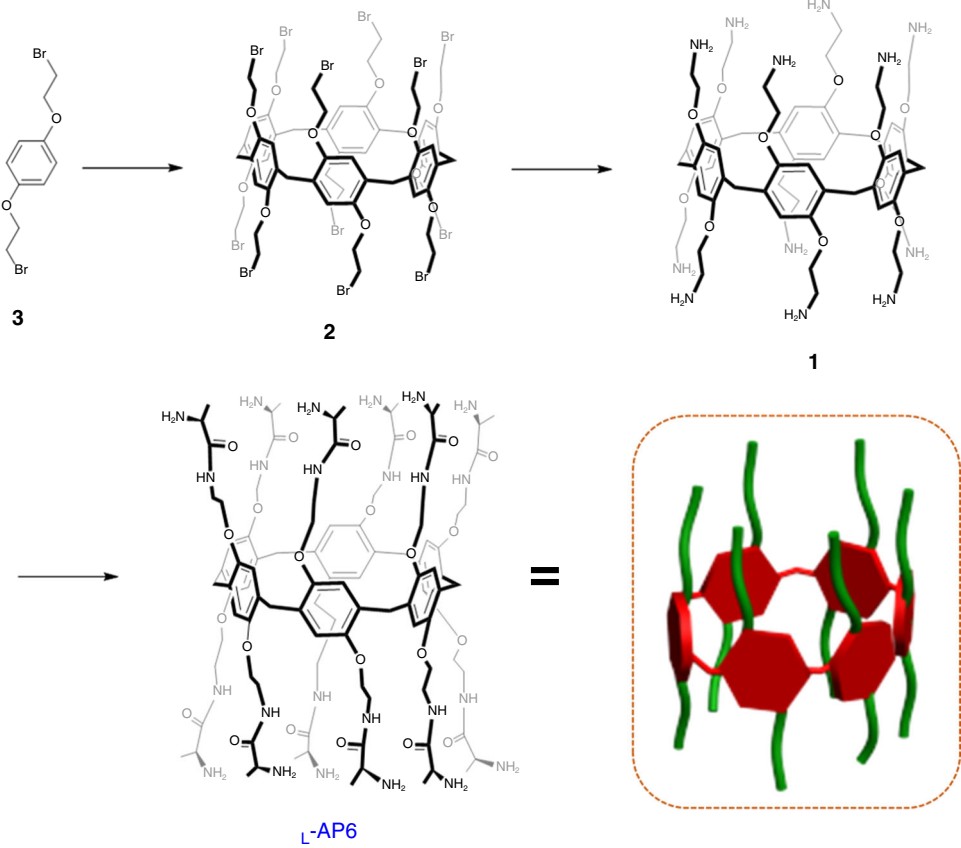

**Fig. 2** Design and high efficient synthesis of chiral pillar[6]arene decorated with L-alanine

intelligent biomimetic nanochannel. First, the PET membrane, treated with heavy ion track technology, can be generated for a single conical nanochannel by chemical etching method (see Supplementary Fig. 4)[34]. As shown in Supplementary Fig. 5, scanning electron microscope (SEM) was used for confirming the structure of the conical nanochannel. The large opening was approximately 580 nm diameter. And the narrow opening at the opposite side was approximately 18 nm diameter. The exposed carboxyl (–COO⁻) groups were produced in the interior walls of the nanochannel during chemical etching process. The amino terminal on AZO connected with the –COO⁻ groups by classical condensation reaction. Additionally, AZO-based nanochannel, immersed in L-AP6 solution, spontaneously assembled into binary complex (detail in Methods). Hence, the internal walls of the nanochannel can be decorated with L-AP6 by a simple two-step reaction. The I–V curves were performed on each modification process (Fig. 3c and Supplementary Fig. 6). Specifically, the original conical nanochannel was negatively charged at neutral condition because of ionized carboxyl groups, and it exhibited strongly rectified ionic current. When AZO was immobilized on the nanochannel, the current decreased, obviously, which is due to the change of charge and wettability on the internal surface. Likewise, the current was further reduced after L-AP6 was assembled into the nanochannel, which can be explained by the changes of the hydrophobic effect and the surface charge of L-AP6. After modification, the current changes indicated that the chiral nanochannel was fabricated successfully.

Both contact angle (CA) measurements and X-ray photoelectron spectroscopy (XPS) experiments were further performed to confirm the successful construction of the biomimetic channel (Supplementary Tables 1–3). The CA of the original membrane bearing inactivated –COO⁻ group was 50.3 ± 1.2° (Supplementary

Fig. 7). The CA of the AZO-modified surface increased to 70.6 ± 1.2°, caused by the hydrophobicity of AZO compound. After the immobilization of L-AP6, the CA continued to increase to 85.2 ± 3.0° owing to the hydrophobicity of L-AP6, which may lead to a decrease in current for the L-AP6-modified nanochannel (Fig. 3d). In addition, the XPS analysis demonstrated that the undecorated PET films did not possess nitrogen element; after modification with AZO, the peak for elemental nitrogen appeared clearly. To characterize the successful assembly of chiral pillararenes and eliminate interference by the nitrogen present in AZO, dansyl chloride labeling pillararenes (L-AP6-DNS) are synthesized according to a literature method as it contained a sulfur element[35]. The $S2p$ peak appears at 163.43 eV after L-AP6-DNS self-assembled, indicating the successful construction of chiral host–guest systems by self-assembly (See Supplementary Fig. 8). Hence, all data emphasize that the L-AP6-based chiral nanochannel is successfully developed by self-assembly strategy.

**A biomimetic chiral-driven ionic gate.** Based on above research, to further confirm the chiral gate of the biomimetic nanochannel, D-Glu/L-Glu is added to the L-AP6-modified nanochannel. The chiral gated performance is characterized by transmembrane ion current in 0.1 M KCl (pH 7.03) while adding 1 mM D-Glu/L-Glu, respectively. When the bare channel and AZO-modified nanochannel are injected with the electrolyte containing D-Glu/L-Glu (1.0 mM), no obvious change in the ionic current is observed (Fig. 4c and Supplementary Fig. 9). However, after immobilization of L-AP6, there is a remarkable difference: the decreasing current was clearly observed in the presence of 1 mM D-Glu; the current remains nearly unchanged in the presence of 1 mM L-Glu (Fig. 4a). The biomimetic chiral-driven ionic gate exhibits both

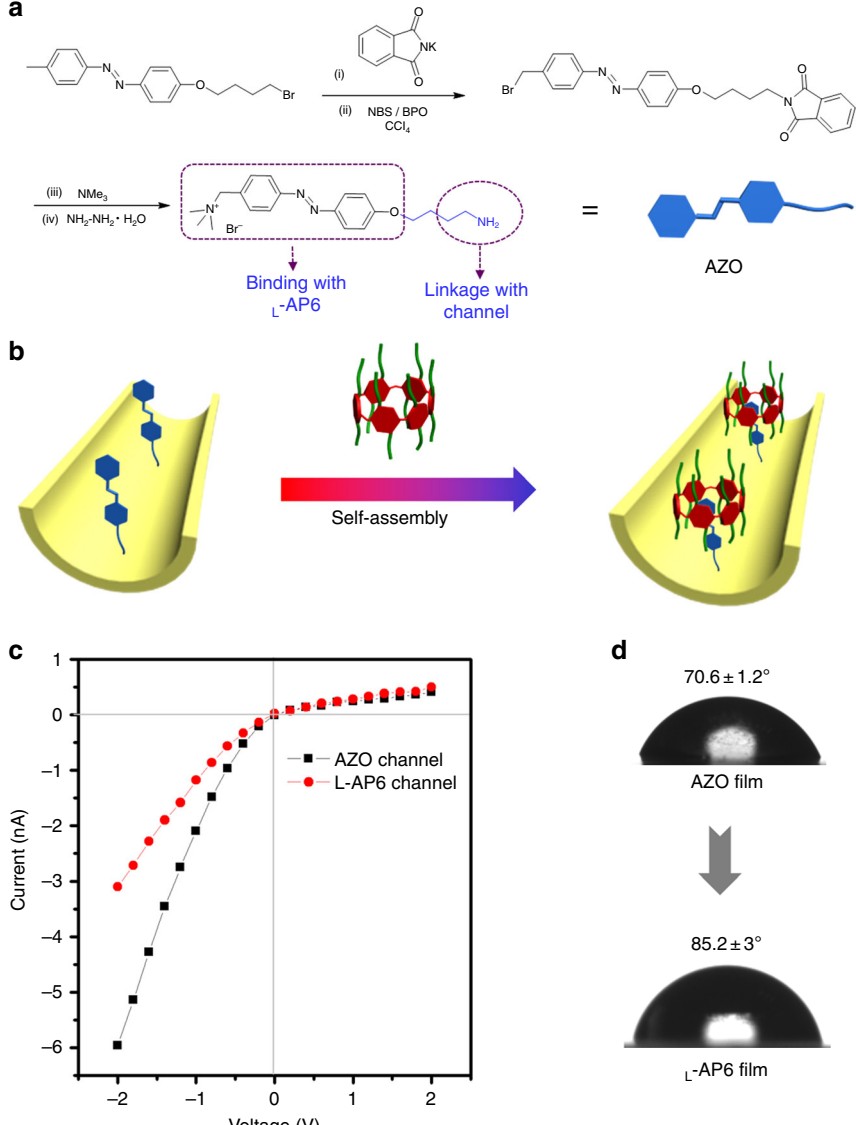

**Fig. 3** Fabrication of the chiral nanochannel. **a** Design and synthesis of the AZO linkage unit; **b** the construction process of the chiral nanochannel by host–guest systems; **c** current/voltage ($I/V$) curve change in the single nanochannel after each modification; **d** contact angle change on the PET films after each modification

switchable ability and stabilized reversibility by the addition and removal of D-Glu. Before adding D-Glu, the L-AP6-modified nanochannel was in the ON-state with a high current level. After activation with D-Glu for 5 min, the L-AP6-modified nanochannel, the transmembrane current changes to −0.5 nA and the L-AP6-modified nanochannel acquired the OFF-state with a lower current level. Followed with immersing the L-AP6-modified nanochannel into pure water for 2 h, most of the D-Glu is released and the system returned to the ON-state. No damping of the ionic current appeared even after several cycles, suggesting that the ionic gate has good repeatability performance in regulating the ion conduction of the nanochannel (Fig. 4b). To more clearly observe the chiral gated performance of the L-AP6-modified channel, a bar graph of the ionic current change ratios is presented in Fig. 4c (the current change ratio is defined as the absolute value, i.e., $I_o/I_{saccharides}$, $I_o$ and $I_{saccharides}$ are the current measured at −2 V before and after treating with saccharides, respectively). Additionally, to outline the validity and scope of this absolutely spectacular result, galactose enantiomers and xylose enantiomers were added to the L-AP6-modified

nanochannels. Surprisingly, the L-AP6-modified channel also exhibits the chiral discrimination performance toward galactose enantiomers and xylose enantiomers, but efficiency ratio less productive than glucose enantiomers (Fig. 4c and Supplementary Figs. 11 and 12). We speculate that it would be possible to describe the binding of saccharide enantiomers on the internal surface of the chiral channel using the Langmuir model (See Supplementary Table 4). And chiral gating performance may be further amplified when the saccharide enantiomers are in the L-AP6-modified nanochannel.

**The mechanism of chiral-driven ionic gate.** To explore the molecular mechanism for the chiral gated channel, we primarily investigate the chiral discrimination performance of the L-AP6-AZO complex toward D-Glu/L-Glu in the solution. The binding of L-AP6-AZO complex to D-Glu/L-Glu was studied by using ${}^1$H NMR titrations (Supplementary Figs. 13 and 14). The signal of proton $H_a$ on L-AP6 exhibited obvious upfield shift as sequential D-Glu/L-Glu additions. These changes in the chemical shifts happened because L-AP6-AZO can immobilize D-Glu/L-Glu via

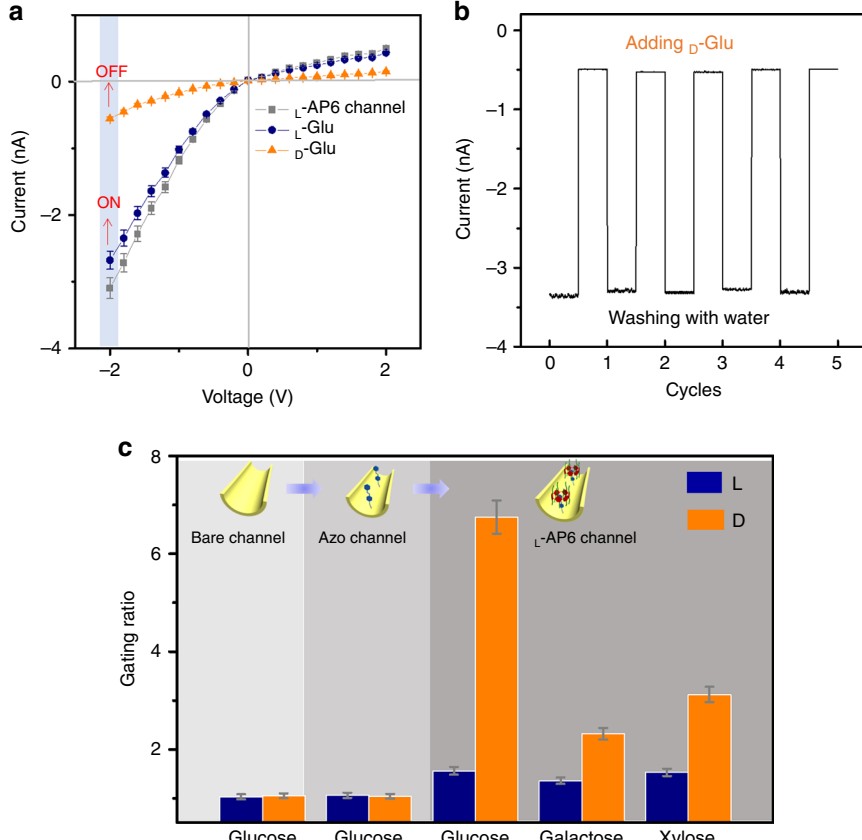

**Fig. 4** A biomimetic chiral-driven ionic gate. **a** *I–V* curves of the L-AP6 nanochannel in 0.1 M KCl electrolyte in the presence of 1 mM glucose enantiomers; **b** responsive switchability of the chiral-driven ionic gate: the variation in the reversible ionic currents of the L-AP6-modified nanochannel measured alternately at a constant voltage of −2 V with the addition and removal of D-Glu; **c** a bar graph of gating ratios $I_o/I_{glucose}$ at −2 V after adding the above saccharide enantiomers in a bare channel, AZO channel, and L-AP6 channel, respectively. Standard deviation is ±5% and is used for describing the error bars. Each dataset in two cases is tested five times, respectively

multiple chiral hydrogen-bonding interactions. Selectivity for D-Glu is greater than ca. 10 times that of affinity of L-AP6-AZO complex for L-Glu. Subsequently, the binding of L-AP6-AZO and D-Glu/L-Glu was also examined by molecular simulation using Gaussian calculation. It indicated that molecular simulations were generally consistent with the $^1$H NMR titrations experiment (see Supplementary Fig. 15). All the evidences clearly demonstrate that the L-AP6-AZO complex shows high chiral preference for D-Glu. And the mechanism for the chiral discrimination is likely based on multiple chiral hydrogen-bonding interactions between the L-AP6-AZO unit and glucose enantiomers.

In the light of above studies, we proposed the potential mechanism of chiral gate of biomimetic glucose-sensitive ion channels. When the L-AP6-modified chiral nanochannels had access to D-Glu solutions, the L-AP6 was capped with D-Glu through the intermolecular hydrogen bond interaction, an appropriate fitting of the functional and geometric requirements, which breaks away the charge from the channel surface, leading to the ionic current decreased significantly (Fig. 5 and Supplementary Fig. 16)[36]. To support the mechanism of glucose-enantiomer-driven ion gate, the surface $pK_a$ of the L-AP6 modified nanochannel upon adsorption of D-Glu/L-Glu can be calculated (detailed in Supplementary methods). As shown in Supplementary Fig. 17, the current at −2 V can be determined and plotted against pH before and after adsorption of glucose enantiomers. The surface $pK_a$ at the neutral condition before adding the D-Glu/L-Glu is approximately 6.08. After binding with glucose enantiomers, the surface $pK_a$ at the neutral condition is approximately 6.71 for

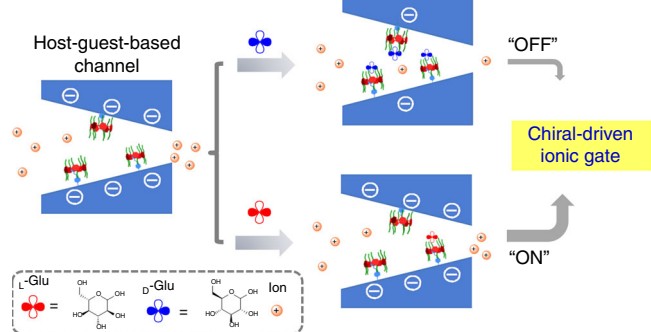

**Fig. 5** The mechanism of glucose-enantiomer-driven ion gate using host–guest systems

binding D-Glu and is approximately 6.14 for binding L-Glu. The difference in surface $pK_a$ in the nanochannels between the binding glucose enantiomers can be explained by the decrease of ionic current after adsorption of D-Glu.

The mechanism was further confirmed by a series of electroosmotic flow (EOF) experiments[37,38]. Phenol, served as an electrically neutral probe, is used for measuring the EOF rate from the feed to the permeate. The diagram related to phenol transport vs. time can be plotted by periodically measuring the fluorescence emission of the phenol in the permeate solution (detailed in Methods). Analogous experiments were completed in the absence of applied current to determine the rate of diffusion

$(N_{\text{diff}})$. $N_i$ (the rate of transport with applied current $i$) and $N_{\text{diff}}$ were used to calculate the enhancement factor (Eq. 1), $E$:

$$E = \frac{N_i}{N_{\text{diff}}}. \tag{1}$$

Peclet number (Pe) can be determined (Eq. (2)) as

$$E = \frac{\text{Pe}}{1 - e^{-\text{Pe}}}. \tag{2}$$

$N_{\text{diff}}$ rate of phenol diffusion and $N_i$ rate of phenol transport in the applied current can be obtained through this process. The relationship between Pe and $V_{\text{eof}}$ is determined using Eq. (3), where $D$ is the diffusion coefficient for phenol and $L$ is the membrane thickness. The diffusion coefficient of phenol ($D = 8.9 \times 10^{-6}$ cm$^2$s$^{-1}$) was used to calculate $V_{\text{eof}}$[38]:

$$V_{\text{eof}} = \frac{\text{Pe}\,D}{L}. \tag{3}$$

The $\zeta$ potential of the nanochannel walls can be determined (Eq. (4)) ($\varepsilon$ and $\eta$ are the permittivity and viscosity of the solution, respectively, $\varepsilon = 6.95 \times 10^{-10}$ c$^2$ J$^{-1}$ m$^{-1}$, $\eta = 2.23$ K$\Omega$; $J_{\text{app}}$ is the constant applied current density; $\rho$ is the resistivity of the electrolyte within the nanochannel). We may extract the $\zeta$ and then surface charge density values can be estimated from the Gouy–Chapman equation ($K^{-1}$ is measured resistivity) (Eq. (5))[37,39].

$$V_{\text{eof}} = \frac{-\varepsilon \zeta J_{\text{app}} \rho}{\eta}, \tag{4}$$

$$\sigma = \frac{\varepsilon \zeta}{\kappa^{-1}}. \tag{5}$$

Hence, the surface charge density can be calculated from above equation. As shown in Fig. 6, with the increasing concentration of D-Glu, the charge of channel surface decreased, mainly due to the formation of a complex between the L-AP6-modifed channel and D-Glu, leading to the ionic current being decreased significantly. From the calculation of surface p$K_a$ and the EOF experiments, it can be illustrated that the reason for the chiral gate in the nanochannel may be the different surface charge densities on the channel walls generated by the binding of D-Glu/L-Glu.

## Discussion

In summary, inspired by glucose-sensitive ion channels, we have described a type of chiral-driven ion gate by introducing chiral pillar[6]arene-based host–guest systems. The functional nanochannel shows high chiral gate for glucose enantiomers and can

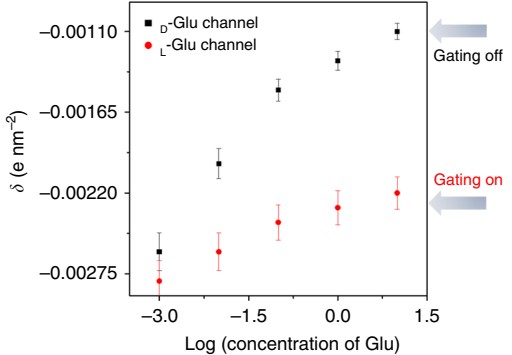

**Fig. 6** The relationship of surface charge density vs. Log (concentration of Glu). Standard deviation is ±5% and is used for describing the error bars. Each dataset in two cases is tested five times, respectively

be switched "off" by D-Glu and be switched "on" by L-Glu. Amazingly, the chiral nanochannel also exhibited a good reversibility toward glucose enantiomers. The chiral gate of the channel may be attributed to the different surface charges generated by the binding of D-Glu/L-Glu. This study not only gives us some insight on the research of biological and pathological processes caused by glucose-sensitive ion channels, but also paves the way for simulating the gating functions of specific channels.

## Methods

**Synthesis of L-AP6.** Compound H2 (100 mg, 0.019 mmol) was reacted with TFA (2 mL) for 5 h. The mixture was evaporated under vacuum to generate L-AP6 (41 mg, 92%). [1]H NMR (400 MHz, D$_2$O): δ 6.59 (s, 12H), 3.68 (s, 24H), 3.65 (s, 12H), 3.33 (s, 12H), 3.29 (d, 24H), 1.02 ppm (d, $J$ = 4.0, 36H) ppm; [13]C NMR (100 MHz, CH$_3$OH-d4): δ 160.50, 160.17, 149.20, 126.83, 116.99, 114.06, 113.80, 65.65, 38.04, 15.15 ppm. MALDI-TOF-MS: calcd for C$_{102}$H$_{150}$O$_{24}$N$_{24}$: 2101.16; found: 2124.23 [M + Na]$^+$. Anal. calcd for C$_{102}$H$_{150}$O$_{24}$N$_{24}$: C, 58.44; H, 7.21, N, 16.03; found: C, 58.55; H, 7.26; N, 16.00.

**Synthesis of AZO.** A solution of G2 (493 mg, 1 mmol) in ethanol (50.0 mL) and trimethylamine (10.0 mL) was added at 80 °C for further reacting for 24 h. Then, hydrazine hydrate was injected into the mixture, to further react for 12 h. The solvents were removed and the residue was washed with dichloromethane to give an orange solid (145 mg, 41%). [1]H NMR (400 MHz, CDCl$_3$): δ 7.82 (m, $J$ = 8.0 Hz, 4H), 7.62 (d, $J$ = 8.0 Hz, 2H), 7.07 (d, $J$ = 8.0 Hz, 2H), 4.46 (s, 2H), 4.10 (d, $J$ = 8.0 Hz, 2H), 3.05 (s, 9H), 1.78 (s, 4H). [13]C NMR (100 MHz, DMSO): δ 166.63, 157.78, 151.08, 138.85, 135.26, 129.91, 127.49, 120.11, 72.62, 59.58, 57.10, 30.83, 29.42. MALDI-TOF-MS: calcd for C$_{20}$H$_{29}$N$_4$OBr: 420.15; found: 341.3 [M-Br]$^+$. Anal. calcd for C$_{20}$H$_{29}$N$_4$OBr: C, 57.01; H, 6.94; N, 13.30; found: C, 56.97; H, 6.98; N, 13.30.

**Nanochannels preparation.** The PET film was used to generate the conical nanochannel using heavy ions track etching technology. Both sides of PET films were irradiated with 365 nm UV light for 1 h before etching process. The PET film was then placed under the electrolytic cells at 30 °C. On one end of the electrolytic cells, etchant solution (9 M NaOH) was injected, while on the other end the stopping solution (1 M KCl+1 M HCOOH) was injected. The ideal tip diameter of the nanochannel can be determined by applying a voltage (1 V) across the device and measuring the ideally resulting current. After etching, the prepared conical nanochannel was placed into MilliQ water for removing the residual salts.

**Nanochannel functionalization.** The functional carboxyl (–COOH) groups were produced in the interior walls of the nanochannel during the chemical etching process. The conical nanochannel was activated into NHS-ester for 1 h at room temperature by immersing in an aqueous solution containing 15 mg EDC and 3 mg NHS. The functional PET film, washing with pure water, was further reacted with azobenzene (5 mM) for 12 h. After that, the azobenzene-based nanochannel, immersed in L-AP6 solution (10$^{-3}$ M) for 5 h, spontaneously assembled into binary complex. The L-AP6-modified chiral nanochannel, further washing with distilled water, was constructed successfully.

**Ion currents measurement.** Keithley 6487 picoammeter (Keithley Instruments, Cleveland, OH) was used to collect the ion currents. Ag/AgCl electrodes were placed on two sides of the electrolytic cells containing PET film with fresh KCl (0.1 M) solution. The recording $I–V$ curves were generated under scanning field signal ranging from −2 to +2 V with a 40-s period. Each experimental dataset was repeated five times to get the average current value.

**CA measurement.** The OCA 20 CA system (Dataphysics, Germany) was used to measure CAs. In case of measuring the CA of the bare film, the original PET membrane was etched by sodium hydroxide solution (9 M) for 1 h. The etched membrane was taken out and treated with stopping solution (1 M HCOOH). The PET membrane was then placed in distilled water for 5 h. The average value of CA was determined by measuring five different positions of the identical PET membrane.

**XPS experiments.** An ESCALab220i–XL electron spectrometer from VG Scientific was used to collect XPS data by using 300 W Al K$_\alpha$ radiation.

**Molecular stimulations.** Molecular stimulations were performed at the density functional theory (DFT) b3lyp/6-31G (d) levels using Gaussian 03.

**EOF experiments.** In case of measuring EOF of the D-Glu channel, one end of the electrolytic cell was injected with the mixture solution (ca. 2 mL) of phenol and D-Glu (10 mM) in KCl electrolyte solution (0.1 M). The other end only contained

electrolyte solution (0.1 M). A fixed voltage was exerted on both sides, namely the permeated solution and the feed solution. A fluorescence spectrometry system was used to measure the fluorescence emission spectrum of the phenol in the permeate solution. The diffusion of phenol across the asymmetric nanochannel without applying a voltage was also measured at the same 30-min interval. The 30-min transport cycles were repeated five times, and the total time we applied current was 150 min.

**Data availability**. The authors declare that the data supporting the findings of this study are available within the article and its Supplementary Information files, and all data are available from the authors on reasonable request.

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

## Acknowledgements

This work was financially supported by the National Natural Science Foundation of China (21572076, 21372092), the 111 Project (B17019), Wuhan scientific and technological projects (2015020101010079).

## Author contributions

Y.S. and H.B.L. conceived and designed the experiments. Y.S. made a major contribution in all experiment. Y.S., F.Z., J.X.Q., F.Z., W.H., J.K.M., H.P., Y.S., D.M.T., and H.B.L. wrote the manuscript. Y.S. and H.B.L. supervised all experiments and analyses.

## Additional information

**Competing interests:** The authors declare no competing interests.

