## [Peer Review File · Nature Communications]

Reviewers' comments:

Reviewer #1 (Remarks to the Author):

This is an interesting paper on nanochannel. In the manuscript, the author reported a new type of chiral glucose-sensitive nanochannel gate which was constructed from the chiral pillar[6]arene-decorated nanopore. The experimental work is at a high standard. I recommend its publication, but I have some important suggestions for revision:

(1) There are many mistakes: Paragraph 2, Page 3: "As depicted in Fig. 2D", Paragraph 1, Page 4: "current change ratios was presented in Figure 3C"

The text should be carefully revised.

(2) The author claims that the channel is biomimetic or bioinspired. However, there is not any element related to the natural channel.

(3) The author claims that the channel gate could be driven by chiral glucose. How about other chiral solutes, such as amino acids? How about the substrate selectivity?

(4) The proposed gating mechanism is too simple to make sense. The interaction mode of glucose with the pillar[6]arene should be described. Did the addition of the glucose lead to the replacement of AZO or bind to the out surface of the pillar[6]arene backbone? More experiments should be conducted.

Reviewer #2 (Remarks to the Author):

The authors report chiral-driven ionic gate system using host-guest complexes between chiral pillar[6]arene and azobenzene complex. The paper is well-written, and glucose chirality can be recognized by the ion gate system. But, preciously, the authors reported ionic channel systems using pillar[5]arene and guest complex. The concept is same, thus this is weak point of the paper. Chiral transportation using chiral pillar[6]arenes has been already reported by Hou et al. (J. Am. Chem. Soc. 2013, 135, 2152). At least, the authors need to cite the paper.

1) Please mention why the authors use chirality of pillar[6]arene. The reviewer thinks that same experiment should be done by introduction of chiral substituents. The planar chirality is important in this system? What happens when the authors introduce asymmetrical carbon (for example, L-alanine) on the surface?

2) Why the authors used pillar[6]arenes instead of pillar[5]arenes?

2) The binding constant between azobenzene and pillar[6]arene was small. The reviewer wondered the stability of the complex.

3) The authors need to check the binding constants in case of Gal and Xyl.

4) Important pillararene references should be included (JACS 2008, 130, 5002, Chem. Rev. 2016, 116, 7937.)

Reviewer #3 made only remarks to the editor and is positive about the paper.

Reviewer #4 (Remarks to the Author):

In this manuscript, the authors have reported a chiral-driven ionic gate that is comprised of a pillar[6]arene-based host-guest system. The obtained results might be interesting as a novel biomimetic functional material and promising. However, it lacks a fully appropriate explanation about a mechanism of the chiral selectivity and also does quantitative analysis. The reviewer feels more elaboration is needed to reach the standard level of the Nature Communication and suggests it to be rejected and resubmitted after detailed analysis on the physical mechanism of them. The queries on the present manuscript are given below.

(1) Concentration dependence of salt and also the AZO linker/Chiral Pillar[6]arenes.

(2) Selectivity of salt (not only metal ions, but also other organic and inorganic cations) is explored.

(3) Similarity and dissimilarity to the glucose dependent ion channel in biological systems should be discussed.

(4) Detailed molecular mechanism on the basis of quantitative analyses is missing. The authors only showed a schematic view in Fig. 5. Is that true? The authors must confirm why the surface charge of nanochannel decreases owing to the adsorption of D-glucose.

Only after the above queries are solved. The review again reconsiders the decision.

Followings are comments on the present manuscript, the authors optionally consider the revision.

(5) At the neutral condition, NH_2 groups are positively charged as NH_3^+ and COO^- groups compensate the positive charge. The reviewer considers that the adsorption of chiral glucose changes the protonation states of amino groups, i.e. carboxyl groups, as indicated in Fig. 5. The authors should confirm the pK_a or pK_b of surface and/or pillar[6]arene upon adsorption of D-glucose. The theoretical analyses on the adsorption mechanism of it is also acceptable. Moreover, the theoretical analyses on the diffusion of the metal ions in charged and non-charged environments might be performed.

(6) When the L-alanines are replaced by D-alanines, one can have an opposite chiral selective system. The authors may confirm the possibility of the L-glucose dependent system, then the present chiral selectivity is truly proven.

(7) pH dependence.

Reviewer #1 (Remarks to the Author):

This is an interesting paper on nanochannel. In the manuscript, the author reported a new type of chiral glucose-sensitive nanochannel gate which was constructed from the chiral pillar[6]arene-decorated nanopore. The experimental work is at a high standard. I recommend its publication, but I have some important suggestions for revision:

(1) There are many mistakes: Paragraph 2, Page 3: "As depicted in Fig. 2D", Paragraph 1, Page 4: "current change ratios was presented in Figure 3C"....

The text should be carefully revised.

(2) The author claims that the channel is biomimetic or bioinspired. However, there is not any element related to the natural channel.

(3) The author claims that the channel gate could be driven by chiral glucose. How about other chiral solutes, such as amino acids? How about the substrate selectivity?

(4) The proposed gating mechanism is too simple to make sense. The interaction mode of glucose with the pillar[6]arene should be described. Did the addition of the glucose lead to the replacement of AZO or bind to the out surface of the pillar[6]arene backbone? More experiments should be conducted.

Response: Thanks for your comments. According to your suggestions, we have done the related experiments.

(1) There are many mistakes: Paragraph 2, Page 3: "As depicted in Fig. 2D", Paragraph 1, Page 4: "current change ratios was presented in Figure 3C"....The text should be carefully revised.

Response: Done.

(2) The author claims that the channel is biomimetic or bioinspired. However, there is not any element related to the natural channel.

Response: In biological organism, the opening and closing of the glucose-sensitive ion channels regulate the transport of ions across cell membranes in response to glucose. Inspired by this, building artificial glucose regulated ion channels in vitro is described. The artificial nanochannels also show glucose-driven ionic gate.

(3) The author claims that the channel gate could be driven by chiral glucose. How about other chiral solutes, such as amino acids? How about the substrate selectivity?

Response: We investigated the \$\text{L}\$ -AP6-modified channel towards the amino acids. As shown in the following figure, the \$\text{L}\$ -AP6-modified channel also exhibited the chiral discrimination towards Alanine enantiomers. Selectivity for \$\text{D}\$ -Alanine is greater than

approximately 1.3 times that of affinity of L-Alanine. However, the channel gating driven by alanine enantiomers is far less effective than towards glucose enantiomers.

Figure 1. (A) I–V curves of the L-AP6 nanochannel in 0.1 M KCl electrolyte in the presence of 1 mM alanine enantiomers; (B) Histogram of gating ratios I_0/I_{alanine} at -2 V after adding the alanine enantiomers in the L-AP6-modified channel.

(4) The proposed gating mechanism is too simple to make sense. The interaction mode of glucose with the pillar[6]arene should be described. Did the addition of the glucose lead to the replacement of AZO or bind to the out surface of the pillar[6]arene backbone? More experiments should be conducted.

Response: We restated the proposed the gating mechanism. The L-AP6 immobilized in the nanochannel via host-guest interactions, inhibit the charge of the nanochannel surface as an $\text{AZO} \cdot \text{L-AP6} \cdot \text{Glucose}$ ternary complex. To support this view, we conducted a series of experiments (The following picture and Yellow parts in manuscript).

We primarily investigate the chiral discrimination performance of the L-AP6-AZO complex toward D-Glu/L-Glu in the solution. The binding of L-AP6-AZO complex to D-Glu/L-Glu was primarily studied by using ^1H NMR titrations (Supplementary information). From the titrations experiments, we may conclude that the L-AP6-AZO can immobilize D-Glu/L-Glu to form the $\text{AZO} \cdot \text{L-AP6} \cdot \text{Glucose}$ ternary complex via multiple hydrogen-bonding sites.

Subsequently, the binding of L-AP6-AZO and D-Glu/L-Glu was also examined by molecular simulation at b3Lyp/6-31G(d) levels by using Gaussian 03. The results from the molecular mechanic calculations were generally consistent with the ^1H NMR spectroscopic experimental results. As shown in the Figure 2, it indicated the L-AP6-AZO complex show high chiral preference for D-Glu.

Figure 2. Energy-minimized complex of _L-AP6 with _L-Glu (left) or _D-Glu (right), optimized at the B3LYP/6–31G* level. This result show that complex (_L-AP6-AZO) prefer to bind _D-Glu.

In the light of above studies, we proposed the potential mechanism of chiral gate of biomimetic glucose-sensitive ion channels. The _L-AP6 immobilized in the nanochannel via host-guest interactions, inhibit the charge of the nanochannel surface as an AZO•L-AP6•Glucose ternary complex. To support the mechanism of glucose-enantiomer-driven ion gate, the surface pK_a of the _L-AP6 modified nanochannel upon adsorption of _D-Glu/_L-Glu can be calculated (Detailed in Supplementary methods). We attempted to calculate the surface pK_a before and after adsorption _D-Glu/_L-Glu. Measuring the current at the -2 V for the couple as a function of pH allows the determination of the pK_a of the surface. The equation commonly used in the buffer region of the acid–base equilibrium is the Henderson–Hasselbach equation:

$$pK_a = pH - \log\left(\frac{[A^-]}{[HA]}\right) \quad (1)$$

The method for determining surface pK_a uses the following argument: if the total current through the electrode is assumed to be composed of two independent parts, one through the dissociated Channel $[A^-]$ and the other through the non-dissociated Channel $[HA]$, then the current can be described by

$$i = i_{A^-}[A^-] + i_{HA}[HA] \quad (2)$$

where i_{A^-} and i_{HA} are currents of the probe on the channel fabricated by $[A^-]$ and $[HA]$

structures, respectively. In this case, [HA] refers to surface concentration, not solution concentration. By setting the channel coverage to 1, the coverage of the surface components is $[A^-] + [HA] = 1$. Using this expression, plus eqs 1 and 2, the following equation can be obtained,

$$pK_a = pH - \log\left(\frac{i_{HA} - i_{A^-}}{i - i_{A^-}} - 1\right) \quad (3)$$

where i_{A^-} and i_{HA} can be determined by the average values of the high pH (> 6) and low pH (< 4), respectively. Hence, a plot of peak current i versus pH can elucidate the pK_a of a surface.

The current at -2 V can be determined and plotted against pH as shown in Figure 4. This plot looks similar to the equivalence part of a titration curve, and using eq 3, the pK_a can be elucidated. Hence, the surface pK_a at the neutral condition before adding the D-/L-Glu is approximately 6.08. After binding with glucose enantiomers, the surface pK_a at the neutral condition is approximately 6.71 for binding D-Glu and is approximately 6.14 for binding L-Glu. The difference surface pK_a in nanochannel between the binding glucose enantiomers can be explained the decrease of ionic current after adsorption of D-Glu.

{Ref: *Journal of Chemical Education*. 82, 779–781 (2005); *Chem. Commun.*, 1338–1339 (2001).}

Figure 4. Relationship between the current at -2 V and the pH value of the solution.

The mechanism in the nanochannel was further confirmed by a series of electroosmotic flow experiments (EOF) (Yellow parts in manuscript) (Figure 5). From

the calculation of surface pKa and the EOF experiments, it can illustrate the reason for the chiral gate in nanochannel, which may be attributed to the different surface charge densities on the channel walls generated by the binding of D-Glu/L-Glu.

Figure 5. Surface charge density vs Log (concentration of Glu). It indicates that the surface charge density decreased gradually with the increasing of concentration.

Reviewer #2 (Remarks to the Author):

The authors report chiral-driven ionic gate system using host-guest complexes between chiral pillar[6]arene and azobenzene complex. The paper is well-written, and glucose chirality can be recognized by the ion gate system. But, preciously, the authors reported ionic channel systems using pillar[5]arene and guest complex. The concept is same, thus this is weak point of the paper. Chiral transportation using chiral pillar[6]arenes has been already reported by Hou et al. (J. Am. Chem. Soc. 2013, 135, 2152). At least, the authors need to cite the paper.

1) Please mention why the authors use chirality of pillar[6]arene. The reviewer thinks that same experiment should be done by introduction of chiral substituents. The planar chirality is important in this system? What happen when the authors introduce asymmetrical carbon (for example, L-alanine) on the surface?

2) Why the authors used pillar[6]arenes instead of pillar[5]arenes?

2) The binding constant between azobenzene and pillar[6]arene was small. The reviewer wondered the stability of the complex.

3) The authors need to check the binding constants in case of Gal and Xyl.

4) Important pillararene references should be included (JACS 2008, 130, 5002, Chem. Rev. 2016, 116, 7937.)

Response: Thanks for your comments. According to your suggestions, we have done the related experiments (detailed response show as follows).

1) Please mention why the authors use chirality of pillar[6]arene. The reviewer thinks that same experiment should be done by introduction of chiral substituents. The planar chirality is important in this system? What happen when the authors introduce asymmetrical carbon (for example, L-alanine) on the surface?

Response: From the CD signal shape, the pillar[6]arene was converted into dual homochirality by directly integrating alanine, thus the CD signals were induced in the absorption region of the \$\pi\$ -conjugated unit. Such multi-homochiral features are highly desirable for enhanced chiral recognition processes that are important for enantioselective discrimination (*Chem. Soc. Rev.*, **2016**, *45*, 3122).

Additionally, we used the \$D_3\$ -AP6 to decorate the channel via the host-guest interactions. As shown in the following picture, it is also worth mentioning that \$D_3\$ -AP6-modified channel showed good chiral selectivity for \$L\$ -Glu.

Figure 1. (A) I–V curves of the D -AP6 nanochannel in 0.1 M KCl electrolyte in the presence of 1 mM glucose enantiomers; (B) Histogram of gating ratios I_o/I_{glucose} at –2 V after adding the glucose enantiomers in the D -AP6-modified channel.

2) Why the authors used pillar[6]arenes instead of pillar[5]arenes?

Response: Comparing with pillar[5]arenes, pillar[6]arene can complex with *trans*-azobenzene owing to the suitable cavity size (*J. Am. Chem. Soc.* **2012**, *134*, 8711).

2) The binding constant between azobenzene and pillar[6]arene was small. The reviewer wondered the stability of the complex.

Response: The complex forming by the azobenzene and pillar[6]arene is stable in nanochannel. To support our claim, the experimental histogram about intensity of current vs. time is given (Figure 2). The functional nanochannel fabricated by pillar[6]arene and azobenzene still formed stable complex with similar current even after a week. This suggests that pillar[6]arene-based host–guest is quite stable in nanochannel.

Figure 2. The stability of the complex between L -AP6 and AZO in the nanochannel.

3) The authors need to check the binding constants in case of Gal and Xyl.

Response: Done.

To investigate the binding constants of Gal enantiomers and Xyl enantiomers, we employed isothermal titration calorimetry (ITC), which is a powerful method for measuring the host-guest association constants (K_a). As shown in Figure 3, the L -AP6 was found to bind L -Gal with low affinity ($K_a < 10 \text{ M}^{-1}$), and the binding constants was approximate 18 M^{-1} with D -Gal. For Xyl enantiomers, the binding constants were approximate 35 M^{-1} with D -Xyl, and approximate 19 M^{-1} with L -Xyl (Figure 4).

Figure 3. The L-AP6 (1.0 mM) with Gal enantiomer (200 mM) in water at 25 °C: (A) with D-Gal ; (B) with L-Gal .

Figure 4. The L-AP6 (1.0 mM) with Xyl enantiomer (200 mM) in water at 25 °C: (A) with D-Xyl ; (B) with L-Xyl .

4) Important pillararene references should be included (JACS 2008, 130, 5002, Chem. Rev. 2016, 116, 7937.)

Response: We have cited the relevant references.

Reviewer #3 made only remarks to the editor and is positive about the paper.

Response: Thanks for your comments.

Reviewer #4 (Remarks to the Author):

In this manuscript, the authors have reported a chiral-driven ionic gate that is comprised of a pillar[6]arene-based host-guest system. The obtained results might be interesting as a novel biomimetic functional material and promising. However, it lacks a fully appropriate explanation about a mechanism of the chiral selectivity and also does not have a quantitative analysis. The reviewer feels more elaboration is needed to reach the standard level of the Nature Communication and suggests it to be rejected and resubmitted after detailed analysis on the physical mechanism of them. The queries on the present manuscript are given below.

- (1) Concentration dependence of salt and also the AZO linker/Chiral Pillar[6]arenes.
- (2) Selectivity of salt (not only metal ions, but also other organic and inorganic cations) is explored.
- (3) Similarity and dissimilarity to the glucose dependent ion channel in biological systems should be discussed.
- (4) Detailed molecular mechanism on the basis of quantitative analyses is missing. The authors only showed a schematic view in Fig. 5. Is that true? The authors must confirm why the surface charge of nanochannel decreases owing to the adsorption of D-glucose.

Only after the above queries are solved. The review will again reconsider the decision. Following are comments on the present manuscript, the authors optionally consider the revision.

- (5) At the neutral condition, NH_2 groups are positively charged as NH_3^+ and COO^- groups compensate the positive charge. The reviewer considers that the adsorption of chiral glucose changes the protonation states of amino groups, i.e. carboxyl groups, as indicated in Fig. 5. The authors should confirm the pK_a or pK_b of surface and/or pillar[6]arene upon adsorption of D-glucose. The theoretical analyses on the adsorption mechanism of it is also acceptable. Moreover, the theoretical analyses on the diffusion of the metal ions in charged and non-charged environments might be performed.
- (6) When the L-alanines are replaced by D-alanines, one can have an opposite chiral selective system. The authors may confirm the possibility of the L-glucose dependent system, then the present chiral selectivity is truly proven.
- (7) pH dependence.

Response: Thanks for your comments.

- (1) Concentration dependence of salt and also the AZO linker/Chiral Pillar[6]arenes.

Response: For low electrolyte concentrations (0.05 M), the gating ratio of glucose enantiomers is approximately 2 for L-Configuration and 4 for D-Configuration, respectively. At the electrolyte concentrations of 0.1 M, the chiral discrimination performance towards L/D -glucose is approximately 2 and 7, respectively. And then with the increasing of electrolyte concentrations, the chiral discrimination decreased, even disappeared at high electrolyte concentrations (1.0 M).

Figure 1. Histogram of gating ratios I_0/I_{glucose} at -2 V after adding the glucose enantiomers in the different concentration of electrolyte.

Additionally, we attempted to increase the concentration of the AZO linker/Pillar[6]arenes to investigate the chiral discrimination performance. With the increasing of chiral pillar[6]arenes, the chiral channel also exhibited the chiral discrimination performance towards glucose enantiomers.

(2) Selectivity of salt (not only metal ions, but also other organic and inorganic cations) is explored.

Response: In our experiment, upon exposure of the L -AP6-modified nanochannel to a solution of D -Glu (1 mM D -Glu in 0.1 M KCl electrolyte solution), selective binding of D -Glu to the channel wall occurred inside the confined geometry. And this effect induced a decrease in the transmembrane ionic current (electrolyte solution, 0.1 M KCl) (testing device is described in Figure 2). Hence, the functionalized nanochannel exhibited good chiral recognition capability toward D -Glu, which was manifested via the changes in the ionic current flowing through the nanochannel.

Figure 2. The schematic diagram for testing L -AP6-modified nanochannel.

(3) Similarity and dissimilarity to the glucose dependent ion channel in biological systems should be discussed.

Response: Similarity: Glucose-sensitive K^+ channels are a class of ionic channels in biological membranes which are blocked by glucose. Inspired by this, we described a type of artificial ion gate by introducing pillar[6]arene-based host-guest systems. The functional nanochannel show high chiral gate for glucose enantiomers. The L -AP6 modified nanochannel can be switched “off” by D -Glu and be switched “on” by L -Glu. Additionally, in terms of shape, both glucose-sensitive K^+ channels and artificial channel are asymmetric.

Figure 3. The shape of the (A) glucose-sensitive channels and (B) artificial channel.

Dissimilarity: The glucose-sensitive ion channels are kinds of proteins channel which existed in biological membranes. The considerable complexity and variability in biological nanochannels severely hinders the practical studies on glucose-sensitive ion gates. Herein we used solid-state synthetic nanochannels. Also unlike the biological channel, we investigated the chirality regulates the gating behavior of glucose-sensitive ion channels.

(4) Detailed molecular mechanism on the basis of quantitative analyses is missing. The authors only showed a schematic view in Fig. 5. Is that true? The authors must confirm why the surface charge of nanochannel decreases owing to the adsorption of D -glucose.

Response: We extracted the surface charge density of nanochannel after binding with

different concentration of D/L -glucose via by a series of electroosmotic flow experiments (EOF). Specifically, the phenol is used as an electrically neutral probe to measure the EOF rate from the feed to the permeate. This was accomplished by periodically measuring the fluorescence emission of the phenol in the permeate solution and making plots of moles of phenol transport vs time (Detailed in method). Analogous experiments were completed in the absence of applied current to determine the rate of diffusion (N_{diff}). N_i (the rate of transport with applied current i) and N_{diff} were used to calculate the enhancement factor (eq 1), E :

$$E = \frac{N_i}{N_{diff}} \quad (1)$$

which was required to determine the Peclet number (eq 2), Pe

$$E = \frac{Pe}{1 - e^{-Pe}} \quad (2)$$

N_{diff} -rate of phenol diffusion and N_i -rate of phenol transport in the applied current can be obtained through this process. The relationship between Pe and V_{eof} is (eq 3), where D is the diffusion coefficient for phenol and L is the membrane thickness (or, equivalently, the membrane thickness). The diffusion coefficient of phenol ($D = 8.9 \times 10^{-6} \text{ cm}^2\text{s}^{-1}$) was used to calculate V_{eof} .

$$V_{eof} = \frac{Pe D}{L} \quad (3)$$

The ζ potential of the nanochannel walls can be determined (eq 4) (ϵ and η are the permittivity and viscosity of the solution, respectively, $\epsilon = 6.95 \times 10^{-10} \text{ C}^2\text{J}^{-1}\text{m}^{-1}$, $\eta = 2.23 \text{ K}\Omega$; J_{app} is the constant applied current density; ρ is the resistivity of the electrolyte within the nanochannel). We may extract the ζ and then surface charge density values can be estimated from the Gouy-Chapman equation (K^{-1} is measured resistivity) (eq 5).

$$V_{eof} = \frac{-\epsilon\zeta J_{app}\rho}{\eta} \quad (4)$$

$$\sigma = \frac{\epsilon\zeta}{\kappa^{-1}} \quad (5)$$

As shown in the following picture, with the increasing concentration of D -Glu, the charge of channel surface decreased which mainly due to the formation of complex between the L -AP6-modified channel and D -Glu through the intermolecular hydrogen bond interaction, leading to the ionic current decreased significantly. To some extent, it can further explain the reason for the chiral gate in nanochannel, which may be

attributed to the different surface charge densities on the channel walls generated by the binding of D/L -Glu.

Figure 4. Surface charge density vs Log (concentration of Glu). It indicates that the surface charge density decreased gradually with the increasing of concentration.

Ref: *J. Am. Chem. Soc.* 132, 2118–2119 (2010); *J. Phys. Chem. C.* 119, 16633–16638 (2015); *Langmuir.* 21, 4680–4685 (2005); *J. Am. Chem. Soc.* 132, 2118–2119 (2010).

(5) At the neutral condition, NH_2 groups are positively charged as NH_3^+ and COO^- groups compensate the positive charge. The reviewer considers that the adsorption of chiral glucose changes the protonation states of amino groups, i.e. carboxyl groups, as indicated in Fig. 5. The authors should confirm the pK_a or pK_b of surface and/or pillar[6]arene upon adsorption of D-glucose. The theoretical analyses on the adsorption mechanism of it is also acceptable. Moreover, the theoretical analyses on the diffusion of the metal ions in charged and non-charged environments might be performed.

Response: We attempted to calculate the surface pK_a before and after adsorption D -Glu/ L -Glu. Measuring the current at the -2 V for the couple as a function of pH allows the determination of the pK_a of the surface. The equation commonly used in the buffer region of the acid–base equilibrium is the Henderson–Hasselbach equation:

$$\text{pK}_a = \text{pH} - \log\left(\frac{[\text{A}^-]}{[\text{HA}]}\right) \quad (1)$$

The method for determining surface pK_a uses the following argument: if the total current through the electrode is assumed to be composed of two independent parts, one through the dissociated Channel $[\text{A}^-]$ and the other through the non-dissociated Channel $[\text{HA}]$, then the current can be described by

$$i = i_{A^-}[A^-] + i_{HA}[HA] \quad (2)$$

where i_{A^-} and i_{HA} are currents of the probe on the channel fabricated by $[A^-]$ and $[HA]$ structures, respectively. In this case, $[HA]$ refers to surface concentration, not solution concentration. By setting the channel coverage to 1, the coverage of the surface components is $[A^-] + [HA] = 1$. Using this expression, plus eqs 1 and 2, the following equation can be obtained,

$$pK_a = pH - \log\left(\frac{i_{HA} - i_{A^-}}{i - i_{A^-}} - 1\right) \quad (3)$$

where i_{A^-} and i_{HA} can be determined by the average values of the high pH (> 6) and low pH (< 4), respectively. Hence, a plot of peak current i versus pH can elucidate the pK_a of a surface.

The current at -2 V can be determined and plotted against pH as shown in Figure 3. This plot looks similar to the equivalence part of a titration curve, and using eq 3, the pK_a can be elucidated. Hence, the surface pK_a at the neutral condition before adding the D-/L-Glu is approximately 6.08. After binding with glucose enantiomers, the surface pK_a at the neutral condition is approximately 6.71 for binding D-Glu and is approximately 6.14 for binding L-Glu. The difference surface pK_a in nanochannel between the binding glucose enantiomers can be explained the decrease of ionic current after adsorption of D-Glu.

{Ref: *Journal of Chemical Education*. 82, 779–781 (2005); *Chem. Commun.*, 1338–1339 (2001).}

Figure 5. Relationship between the current at -2 V and the pH value of the solution.

(6) When the L-alanines are replaced by D-alanines, one can have an opposite chiral selective system. The authors may confirm the possibility of the L-glucose dependent system, then the present chiral selectivity is truly proven.

Response: According to your suggestions, we used D -AP6 (Pillar[6]arene decorated with D -Alanine) to fabricate chiral nanochannel. As shown in the following picture, it is also worth mentioning that D -AP6-modified nanochannel showed good chiral selectivity for L -Glu.

Figure 6. (A) I–V curves of the D -AP6 nanochannel in 0.1 M KCl electrolyte in the presence of 1 mM glucose enantiomers; (B) Histogram of gating ratios I_o/I_{glucose} at -2 V after adding the glucose enantiomers in the L -AP6-modified channel.

(7) pH dependence.

Response: The L -AP6 modified nanochannel was investigated towards different pH value. As can be seen in the following picture, the functional nanochannel still exists chiral-driven ionic gate property in the normal range of pH value (4.0–9.0).

Figure 7. Relationship between the current at -2 V and the pH value of the α -AP6 nanochannel in 0.1 M KCl electrolyte in the presence of 1 mM glucose enantiomers.

REVIEWERS' COMMENTS:

Reviewer #1 (Remarks to the Author):

I have checked again on the questions of reviewer #2 and other reviewers and the point-by-point response of the authors. The new experiments and explanations can thoroughly respond to the questions of all reviewers. I support its publication.

Reviewer #4 (Remarks to the Author):

In the revised manuscript, the authors have correctly revised the points that the reviewer asked before. Thus, the article is now recommended to publication in Nature Communication as is. One possible refinement in the binding energy analysis by the reviewer is that the author use B3LYP+D3 with PCM to take both van der Waals interaction and solvent effects into account.

Reviewer #4 (Remarks to the Author):

In the revised manuscript, the authors have correctly revised the points that the reviewer asked before. Thus, the article is now recommended to publication in Nature Communication as is. One possible refinement in the binding energy analysis by the reviewer is that the author use B3LYP+D3 with PCM to take both van der Waals interaction and solvent effects into account.

Response: According to your suggestions, we supplemented a top view image of molecular calculations. The intermolecular interactions between chiral L -AP6 and glucose enantiomers are marked (detailed response is described in Supplementary Figure and Supplementary Method).

Supplementary Figure. The top view image of molecular calculations